# Effect of Print Orientation on the Tribological Behavior of a Steel Powder-Modified Thermoplastic

**DOI:** 10.3390/ma19010132

**Published:** 2025-12-30

**Authors:** Krystyna Radoń-Kobus, Paweł Szczygieł

**Affiliations:** Department of Mechatronics and Mechanical Engineering, Kielce University of Technology, al. Tysiąclecia Państwa Polskiego 7, 25-314 Kielce, Poland; pszczygiel@tu.kielce.pl

**Keywords:** 3D printing, mechanical properties, tribological properties

## Abstract

This article presents the results of research on a composite filament made of a thermoplastic polymer with the addition of steel powder, used to produce samples using Fused Deposition Modeling (FDM) 3D printing technology. Samples were printed with different print orientations (0° and 90°) to assess the effect of print direction on mechanical and tribological properties. Sample hardness was tested using the Shore D method. Wettability was determined by measuring the contact angle using an optical tensiometer. Tribological wear tests were conducted using the ball-on-disk method. During the tests, the friction coefficient was recorded, and the wear traces were analyzed using an optical microscope. Friction-wear tests were conducted under dry friction conditions and with a physiological saline solution. The obtained results allowed for determining the relationship between print orientation and the mechanical properties and wear resistance of the analyzed composite material.

## 1. Introduction

In recent years, additive manufacturing technologies have become one of the most popular and important technologies that facilitate the rapid and inexpensive production of components with complex geometries, individually tailored to specific properties and personalized [1,2,3]. Among the wide range of additive manufacturing techniques, Fused Deposition Modeling (FDM) stands out. Its main advantages include low operating costs and high availability of polymer or composite filaments [4]. Despite the relatively intensive development of this technique and the growing interest in it from industry, there is still a lack of in-depth research on its surface and tribological properties. The main disadvantage of FDM is its strong connection with process parameters (extrusion and bed temperature, path filling or orientation, layer thickness or printing speed). They influence not only the internal structure, but also many of their properties, such as mechanical, tribological and surface quality [5,6]. The main characteristic of the resulting samples is the anisotropy of their properties and the resulting interlayer discontinuities. This indicates that studying the wear resistance of materials manufactured using this technology is a key element in understanding the mechanisms involved. This will lead to a more detailed understanding of their operational durability. To improve the quality and reliability of components manufactured using FDM technology, it is crucial to examine their tribological properties under various environmental conditions. The fundamental issue is the simulation of dry friction conditions, with specific and constant parameters such as temperature and humidity. The next step in the research should be testing under lubrication conditions with various lubricants. For several years, manufacturers have been competing to develop various materials used in FDM technology [7]. These include base polymers: Polylactic Acid (PLA), Akrylonitryl-Butadien-Styren (ABS), Poliamid (PA), modified materials (e.g., with nanomaterials or various admixtures, such as glass or carbon fibers), hybrid composites, and even polymers that work on their own to improve their functionality [8,9,10,11]. The use of such a wide range of filaments results in the possibility of significantly improving their wear resistance [12,13]. The essence of introducing new materials to global markets should be their detailed testing in terms of their functional properties under various operating conditions, loads or depending on the geometry of the obtained samples.

Many researchers are analyzing the impact of FDM technology parameter variability on the operational properties of manufactured materials. They are examining the influence of infill level, printing speed, and layer thickness on the variability of mechanical or tribological properties [14,15]. This is definitely important for economic reasons (less material used, lower electricity consumption).

Despite numerous studies on the tribological properties and mechanical strength of components manufactured using FDM technology, research on the latter and the wear mechanisms occurring during their use are still insufficiently investigated [16,17]. There is a lack of consistency in the methodology, surface preparation, and conditions of tribological tests. It is essential to introduce a uniform method for assessing the resulting layers into articles to determine the repeatability of the obtained results.

FDM technology is widely used in various industries (automotive, aerospace, various aspects of mechanical engineering and mechatronics), medicine, and dentistry. Using additive manufacturing, we produce prototypes of housings, handles, machine components, drone and robot components, lightweight auxiliary tools in the automotive industry, and components tested in wind tunnels [18,19,20]. An important feature of three-dimensional (3D) printing that has changed modern medicine is the ability to create individual orthoses, prostheses or surgical instruments to order for a specific patient [21].

The main objectives of this article include an in-depth characterization of the tribological properties of a thermoplastic composite with the addition of steel powder obtained by the FDM method. The main research aspects include the influence of the layer structure, process parameters, type of materials and the load and lubricant used during the tribological test. 

## 2. Materials and Methods

The samples were made from a commercially available material called AIB METAL (AIB, Knurów, Poland). This material is produced in filament form, i.e., a thermoplastic strand with a diameter of 1.75 mm wound on a spool. The manufacturer states that elements printed with this filament resemble steel in appearance and weight. They also boast high density (approximately 4 g/cm^3^) and very good resistance to environmental factors [22]. The filament is a composite in which 316L steel powder is embedded in a thermoplastic polymer. According to information obtained directly from the manufacturer, the steel content in the material is approximately 95% by weight. The exact composition of the filament remains a company secret, but the product data sheet indicates that the material consists of a mixture of polymers and a metallic alloy uniformly dispersed in a polymer matrix in powder form. The data sheet also presents the chemical composition of the steel powder (wt. %), which is similar to standard 316L stainless steel: a dominant iron content (>60%), 16–18% chromium, 10–14% nickel, 2–3% molybdenum, up to 2% manganese, and up to 1% silicon, with a very low carbon content of less than 0.03% [23].

The material parameters are presented in Table 1. According to the manufacturer, these were determined on samples produced using the following settings: layer height of 0.12 mm, 100% fill, linear fill pattern, three contours, nozzle temperature of 240 °C, and bed temperature of 80 °C [24].

Sample preparation began with modeling in CAD (Computer Aided Design) software. The samples were developed based on the ISO 527 standard (sample type 1BA). 3D models of individual components were prepared in SOLIDWORKS 2023 (Dassault Systèmes, Vélizy-Villacoublay, France) and then saved in STL (Standard Tessellation Language) format. After importing the STL files into Bambu Studio software (Bambu Lab, Shenzhen, China), the additive process parameters were configured.

A BambuLab X1C 3D printer (Bambu Lab, Shenzhen, China), operating in FDM (Fused Deposition Modeling) technology, was used to produce the samples. The device was equipped with a hardened steel nozzle with a diameter of 0.6 mm and a textured Polyetherimide (PEI) plate build platform. Two types of samples were prepared depending on the intended use:(a)S1—samples for surface geometric structure analysis and hardness test;(b)S2—cylinders with a diameter of 40 mm and a height of 6 mm intended for surface wettability measurements and tribological test.

The additive manufacturing process parameters are summarized in Table 2. The only variable parameter was the orientation of the samples on the 3D printer build platform, considered at two levels: 0° and 90°, i.e., the angle between the sample surface and the build table surface. Additionally, in the case of S1 samples, tribological tests were conducted under two different conditions: dry friction (DF) and friction with lubricant NaCl 0.9% water solution (0.9NaCl), using countersamples (6 mm diameter balls) made of two different materials: 100Cr6 steel and polyoxymethylene (POM).

The samples were produced in varying numbers of repetitions, as summarized in Table 3 along with their labels. The arrangement of the samples on the 3D printer build platform is shown in Figure 1.

The countersamples used in the tribological tests were 6 mm diameter balls made of two kinds of materials: 100Cr6 steel (AISI 52100) and an unmodified POM (polyoxymethylene) polymer, marketed under the Delrin trademark.

100Cr6 steel balls (low alloy martensitic chrome steel) are characterized by high hardness, wear resistance, excellent surface quality, and dimensional precision. These properties make them widely used in the production of mechanical components such as bearings, shafts, bushings, and other parts subjected to high loads and abrasion. The material demonstrates resistance to high mechanical stresses while maintaining dimensional stability in harsh operating conditions. 100Cr6 steel has the following chemical composition (wt. %): carbon content (%C) ranges from 0.93 to 1.05%, ensuring high hardness and strength. Silicon (%Si) is present in amounts of 0.15–0.35%, which improves the steel’s strength and formability during heat treatment. Manganese (%Mn) in amounts of 0.25–0.45% increases hardenability and wear resistance. Phosphorus (%P) and sulfur (%S) are present in amounts of up to 0.025%, which reduces brittleness and improves mechanical properties. Chromium (%Cr) in the steel is 1.40–1.65%, which improves corrosion and wear resistance. Nickel (%Ni) up to 0.30% and molybdenum (%Mo) up to 0.08% increase strength and temperature stability. Copper (%Cu) is present in amounts of up to 0.20%, contributing to improved resistance to atmospheric corrosion [26]. Properties are listed in Table 4.

The countersamples used in the tribological tests were 6 mm diameter balls made of 100Cr6 (AISI 52100) steel and an unmodified POM (polyoxymethylene) polymer, marketed under the trade name Delrin. These very lightweight POM balls are characterized by good mechanical properties, resistance to corrosion, wear, and abrasion (Table 5). This material is resistant to contact with bases, neutral substances, mild acids, seawater, petroleum products, mineral oils and greases, inorganic salt solutions, aliphatic and aromatic hydrocarbons, chlorine, low-grade alcohol, and ether. However, it is not resistant to strong acids (e.g., hydrochloric, phosphoric, nitric, and sulfuric), some mineral acids, chlorides, and strong bases. Thanks to these properties, POM is used in the food, chemical, electronics, and pharmaceutical industries, including the production of spray mixers, lightweight safety valves, bearings, specialized pumps and valves, fluid flow control devices, and medical instruments [27]. The properties are presented in Table 5.

The hardness was measured using a Shore hardness tester (Hildebrand, Wendlingen, Germany) with a steel cone-shaped indenter with an angle of 30° and a rounded end with a radius of 0.1 mm. Figure 2 shows the hardness measurement procedure. The sample thickness was 6 mm. The ambient temperature was 22 ± 1 °C. The measurement was performed in accordance with the ISO 868 standard for measuring the hardness of plastics and materials of medium to high hardness. The reading was taken after 1 s. Five repetitions were performed for each sample. The research results are presented in Section 3.1.

Surface wettability was measured using an optical tensiometer (Attension Theta Flex, Espoo, Finland). Figure 3 shows how to measure the contact angle. Two liquids were used for the measurements: distilled water as the measuring liquid and 0.9NaCl solution as the tribological test liquid. Five measurements were performed for each pair and with each liquid. The research results are presented in Section 3.2.

Tribological tests were performed on a TRB^3^ tribometer (Anton Paar, Baden, Switzerland) in rotational motion. 40 mm diameter disks in 0° and 90° orientations were used as samples, and 6 mm diameter 100Cr6 steel and POM balls were used as countersamples. The tests were performed under dry friction conditions and friction using a 0.9NaCl solution. Five measurements were performed for each friction pair. Figure 4 shows the friction pairs under both conditions. Table 6 presents the parameters of the tribological tests. The research results are presented in Section 3.3.

The analysis of wear traces after the friction-wear tests was performed using a confocal microscope with interferometric mode DCM8 (Leica, Heerbrugg, Switzerland). A 20× confocal objective was used for the tests. Following the tests, differences between the maximum depths and volumes of the tested surfaces were compared. Figure 5 shows the measurement procedure using the 20× objective. The obtained results are summarized and compared in Section 3.4.

## 3. Results

### 3.1. Hardness Measurement

Figure 6 summarizes the values obtained for hardness tests using the Shore D method.

The results of the Shore D hardness measurements for the 0° and 90° printing angles were 24.1 and 24.2. These were averages of five measurements. The deviation for both samples was 0.05. The difference between the results is small, at 0.4%, and therefore not considered significant. The obtained values (low standard deviation) confirm the repeatability of the printing process. They also indicate that sample orientation during printing has no significant impact on the obtained test results.

### 3.2. Contact Angle

Figure 7 shows sample images of distilled water and 0.9NaCl salt solution droplets deposited on the tested surfaces. Figure 8 summarizes the average contact angle values.

Surface wettability studies indicated a clear dependence of surface properties on print orientation and the type of measuring fluid used. Mean contact angle values are characterized by low standard deviations. Both print angles, 0° and 90°, are hydrophobic surfaces. The distilled water contact angle of the 90° print is approximately 5% higher than that of the 0° print. The contact angles with the 0.9NaCl salt solution are also quite similar, but the 90° print also has a greater contact angle than the 0° print, by approximately 8%.

Polymers are typically characterized by lower surface free energy compared to metallic materials. This results in higher water contact angles. The obtained contact angle values were closer to those commonly reported for polymer surfaces than for metal surfaces. The relatively low surface energy resulting from contact angle measurements supports the conclusion that the polymer phase is located on the surface. The lack of a low contact angle (around 70°) directly supports the claim that polymer surfaces are superior to metal surfaces (regardless of the addition of steel).

Thus, the results indicate the dominance of the polymer phase on the surface. The 0.9NaCl solution is characterized by lower surface energy. Due to the addition of steel, the material has increased sensitivity to liquids with lower surface energy. For this reason, the contact angles with the 0.9NaCl solution are lower than the contact angles with distilled water. The obtained results confirm that the behavior of this material (obtained contact angle values) is typical for hybrid materials.

### 3.3. Tribological Tests

Figure 9 and Figure 10 show the average values of friction coefficients recorded during dry friction and friction with 0.9NaCl solution for a steel ball (100Cr6) and a plastic ball (POM).

The friction coefficients observed for 0° and 90° printing angles during dry friction were the same. After introducing the 0.9NaCl, they significantly decreased for both cases, by approximately 16% for the 0° angle and by approximately 13% for the 90° angle, respectively. One reason for this relationship (in case of using 0.9NaCl) may be the formation of a thin water film on the surfaces, partially separating the surfaces and reducing the adhesion occurring in the metal-to-metal relationship. The formation of a smooth tribochemical layer is favored by the presence of ions. These include iron and chromium oxides, which act as a natural lubricant. Another advantage of the solution used is the reduction in wear products, which consequently results in a reduction in the abrasive component in friction.

The coefficients of friction observed for 0° and 90° print angles during dry friction differed. The 90° print had a higher coefficient of friction than the 0° print by approximately 11%. A difference was also observed when using the 0.9NaCl solution. The 90° print angle had a significantly higher coefficient of friction by approximately 17%. After introducing the 0.9NaCl solution, the coefficients of friction increased significantly for both print angles by approximately 7% and for the 90° angle by approximately 13%, respectively. This may be due to the lack of a stable lubricating layer. One reason was the significant increase in adhesion forces between the print surface and the POM balls. Another reason is the leaching of the wear product layer, which in this friction pair served as a separator between the surfaces. Cl^−^ ions can also adversely affect samples with 316L steel additives.

### 3.4. Measurements of Wear Traces After Friction-Wear Tests

Figure 11 and Figure 12 present isometric views and surface profiles after friction-wear tests using a steel ball, for dry friction, and for friction with a physiological saline solution. Table 7 compares the average values of the maximum wear depth and surface area of the tested materials.

Analysis of the wear scars during dry friction indicated significantly greater wear at a 0° print angle than at a 90° print angle. The wear scar depth was 2.5 times greater and the wear area was approximately 3.5 times greater. This indicates strong wear anisotropy. A stronger increase in wear was observed for both orientations when rubbing with a 0.9NaCl solution. For 0°, the increase was approximately 5% (depth) and 2% (wear area), and for 90°, approximately 91% (depth) and 143% (wear area), compared to dry friction. The highest wear with a steel ball was observed for the 0° orientation using a 0.9NaCl solution. The analyzed fluid changes the contact conditions by increasing adhesion and enabling particle detachment from the material, resulting in increased surface damage. It is also possible that the electrolyte may contribute to tribocorrosion for both, and consequently, progressive damage to the polymer matrix. The presence of 316L steel particles may result in the formation of local zones with altered mechanical properties and adhesion.

Figure 13 and Figure 14 present isometric views and surface profiles after friction-wear tests using a POM ball, for dry friction and friction with a 0.9NaCl. Table 8 compares the average values of maximum wear depth and surface area for the tested materials.

After friction-wear tests using a POM ball, it was observed that the maximum depth for a 90° print angle was approximately 22% higher than for a 0° print angle under dry friction conditions. The abrasion trace fields are very similar, indicating a lack of strong anisotropy. In the 0.9NaCl environment, the maximum depth of abrasion marks in both orientations decreases significantly compared to dry friction. For 0°, this decrease is approximately 38%, and for 90°, approximately 47%. This may suggest that the 0.9NaCl solution environment reduces the intensity of local penetration in contact with the POM ball. The wear areas behave differently for the two orientations. For 0°, it decreases by approximately 38%, and for 90°, it increases by as much as 41%. This indicates a wider but much shallower abrasion mark for a 90° printing angle. Compared to friction tests using a steel ball, significantly less wear was observed. It is possible to produce so-called polymer transfer using POM. This reduces adhesion, which in turn leads to a reduction in damage depth, especially in solutions. In summary, the influence of the material from which the countersample is made is a key aspect in tribological testing.

This study has several limitations that should be noted. These are due to its preliminary nature. Consideration should be given to a broader range of loads, counter-sample materials, and lubricants used. This will improve the ability to replicate real-world conditions encountered in the human body or during surgery. Future goals also include expanding the analysis to include a wider variety of printing parameters (layer height or printing speed). Future work may expand the scope of the topics discussed to include those mentioned above.

## 4. Conclusions

The tribological tests and surface characterization allowed us to determine the key factors influencing the operational resistance of the tested composite containing 316L steel.

1. The obtained results of the material hardness test indicate that the orientation of the material print does not have a significant effect on hardness.

2. Surface wettability tests demonstrated the hydrophobic nature of the print in both orientations, as evidenced by contact angle values exceeding 90°. This value may be due to the dominance of the polymer phase in the outermost surface layer, which is characterized by low surface free energy (higher contact angle with distilled water).

3. During dry friction and friction in the presence of a 0.9NaCl solution using a 100Cr6 ball, a decrease in the friction coefficient was observed for both print orientations. The reduction reached approximately 13–16% when the 0.9NaCl solution was applied. During dry friction and friction with a 0.9NaCl solution and a POM ball, an increase in the friction coefficients for both print orientations was observed, by approximately 7–13% with the 0.9NaCl solution.

4. Surface analysis after tribological tests with a steel ball showed higher wear at a 0° angle than at a 90° angle under dry friction conditions, and the opposite situation (higher wear at a 90° angle than at 0°) under friction conditions with a 0.9NaCl solution. Surface analysis after tribological tests with a POM ball showed higher wear at a 90° angle than at a 0° angle under dry friction conditions, as was the case under friction conditions with a 0.9NaCl solution.

5. The electrolyte can also initiate tribocorrosion and the gradual destruction of the polymer matrix, while the presence of 316L steel particles promotes the formation of localized areas with altered mechanical and adhesive properties. The 0.9 NaCl solution environment influences the nature of wear, reducing the depth of local penetration in contact with the POM ball, while simultaneously varying the wear area depending on the printing orientation. Compared to the steel countersample, significantly less wear is observed, which may be attributed to the formation of a transfer layer of the POM polymer that limits adhesion. This emphasizes the crucial importance of the countersample material in tribological testing.

## Figures and Tables

**Figure 1 materials-19-00132-f001:**
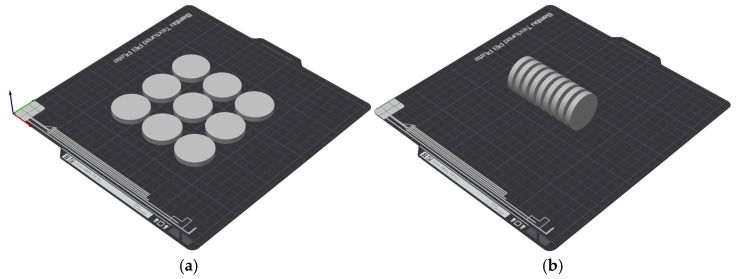
Arrangement of samples on the 3D printer build platform (**a**) S1, S2-1; (**b**) S1, S2-2.

**Figure 2 materials-19-00132-f002:**
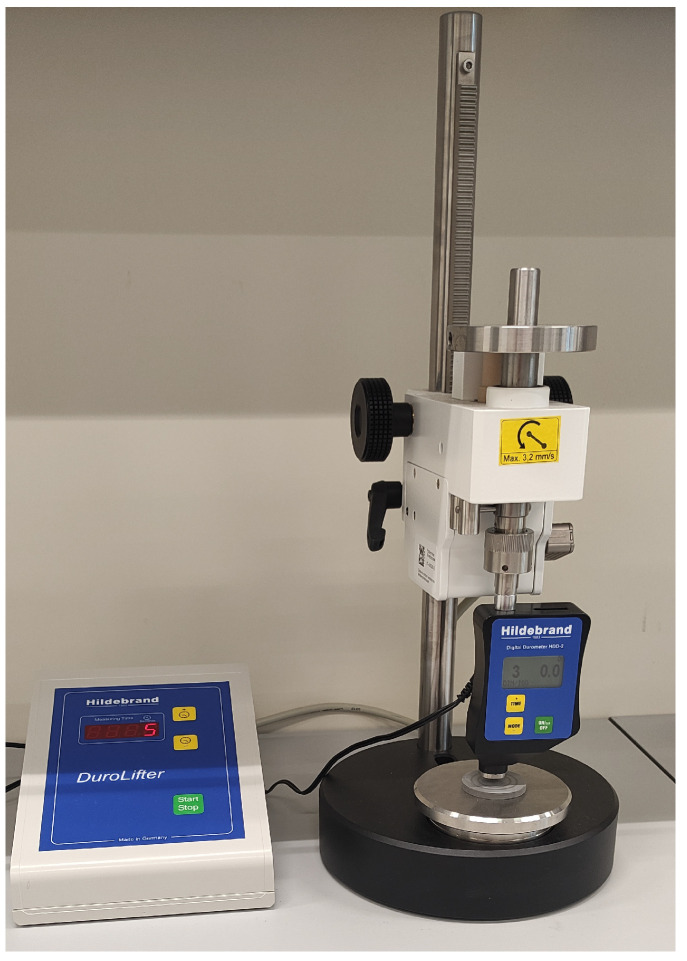
Shore hardness tester.

**Figure 3 materials-19-00132-f003:**
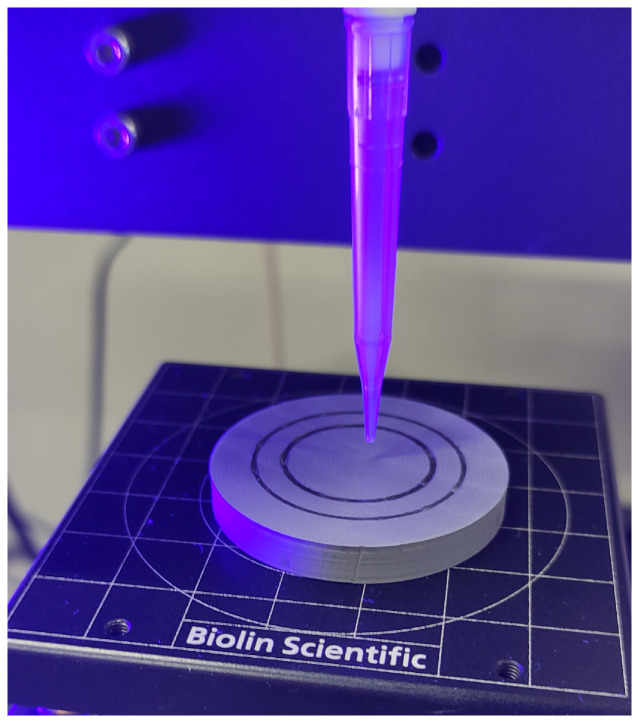
Contact angle test.

**Figure 4 materials-19-00132-f004:**
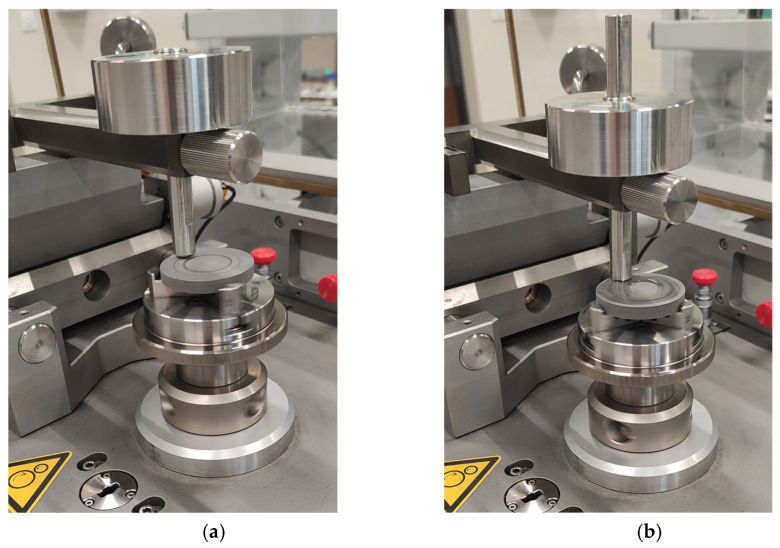
Friction pair (**a**) dry friction; (**b**) friction with 0.9NaCl.

**Figure 5 materials-19-00132-f005:**
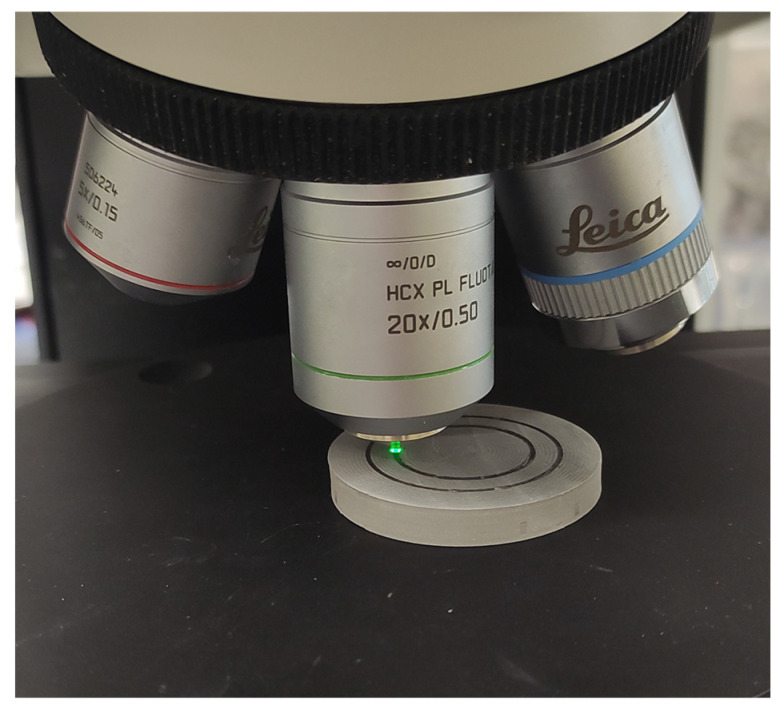
Measurement of wear traces.

**Figure 6 materials-19-00132-f006:**
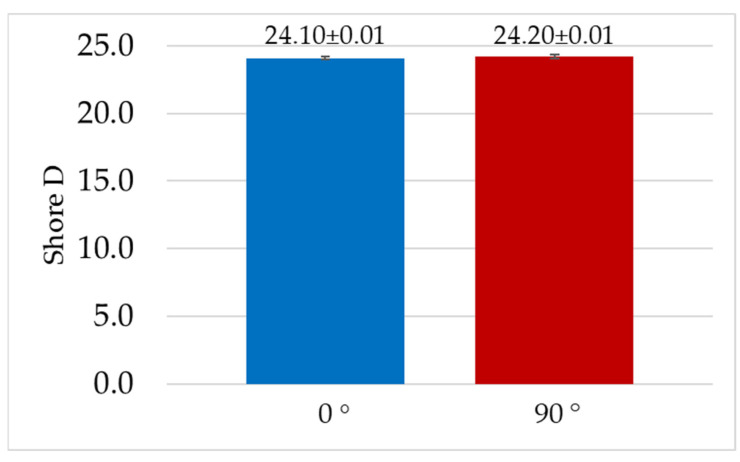
Shore hardness measurements.

**Figure 7 materials-19-00132-f007:**
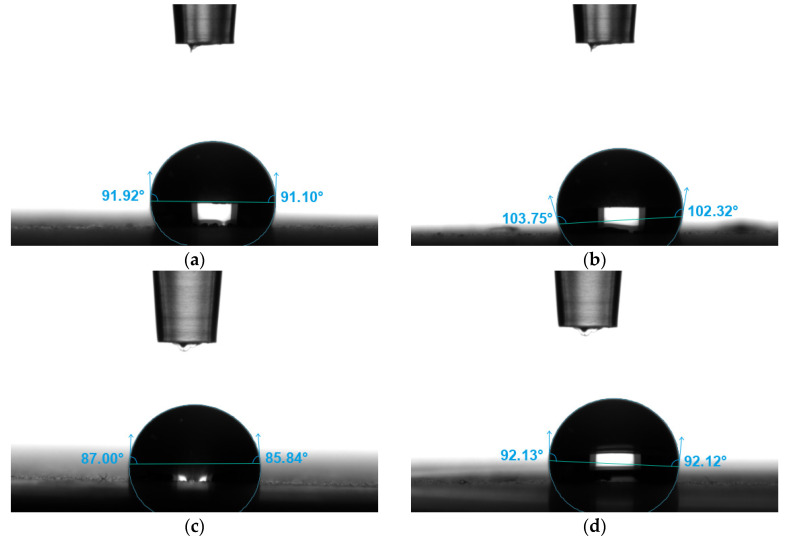
Examples of views of drops of demineralized water (**a**) 0°; (**b**) 90° and 0.9NaCl solution; (**c**) 0°; (**d**) 90°.

**Figure 8 materials-19-00132-f008:**
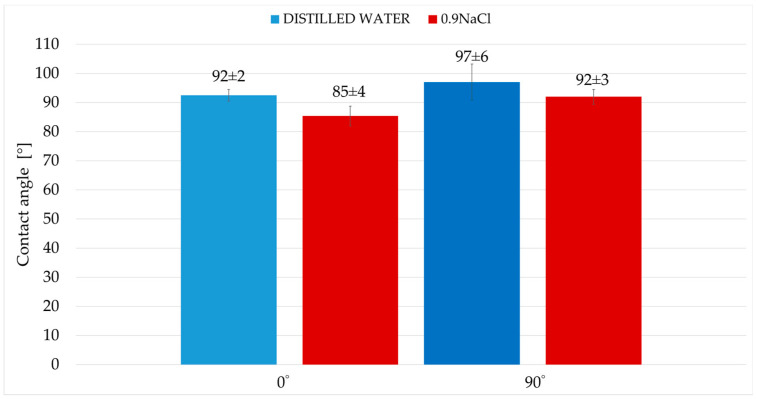
Average contact angle.

**Figure 9 materials-19-00132-f009:**
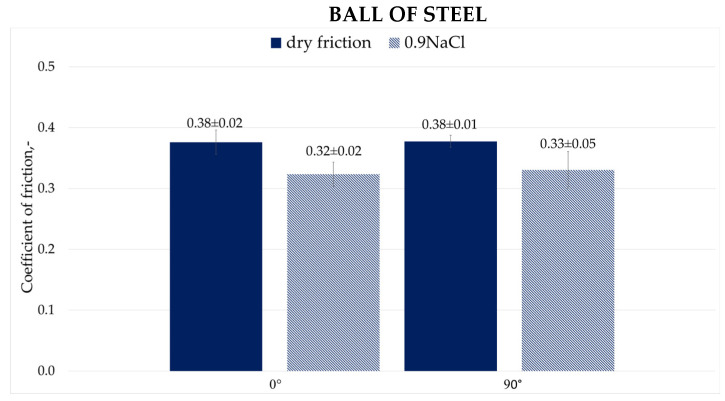
Coefficient of friction for friction with ball of steel.

**Figure 10 materials-19-00132-f010:**
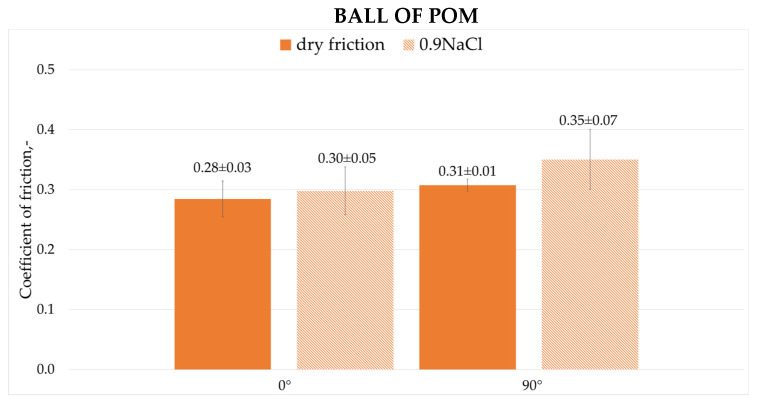
Coefficient of friction for friction with ball of POM.

**Figure 11 materials-19-00132-f011:**
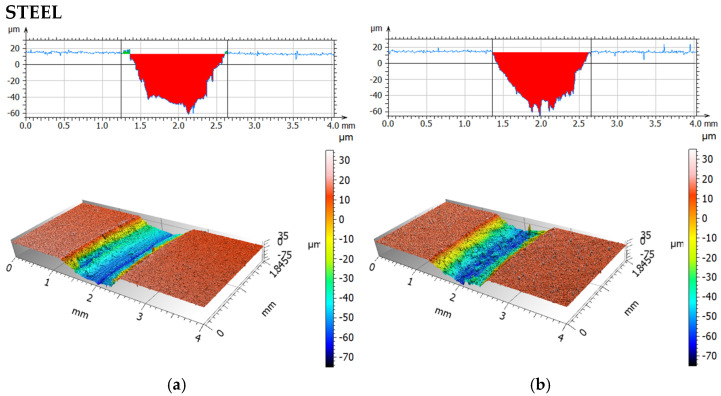
Isometric photos and examples of surface profiles after dry friction (**a**) and with 0.9NaCl; (**b**) for print angle 0° (steel).

**Figure 12 materials-19-00132-f012:**
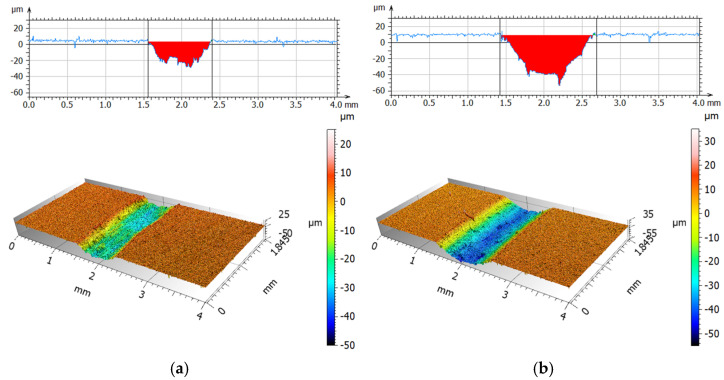
Isometric photos and examples of surface profiles after dry friction (**a**) and with 0.9NaCl; (**b**) for print angle 90° (steel).

**Figure 13 materials-19-00132-f013:**
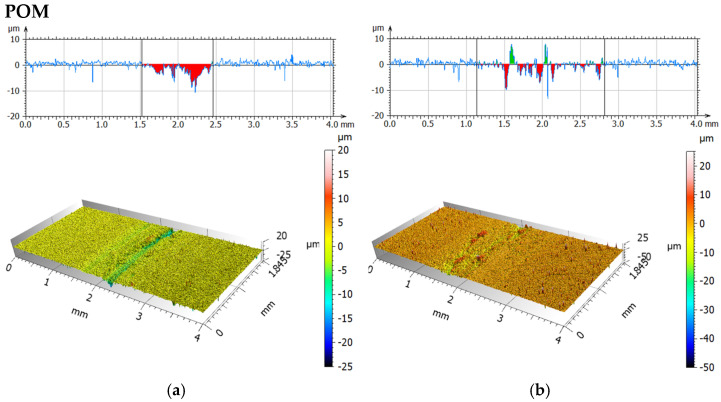
Isometric photos and examples of surface profiles after dry friction (**a**) and with 0.9NaCl; (**b**) for print angle 0° (POM).

**Figure 14 materials-19-00132-f014:**
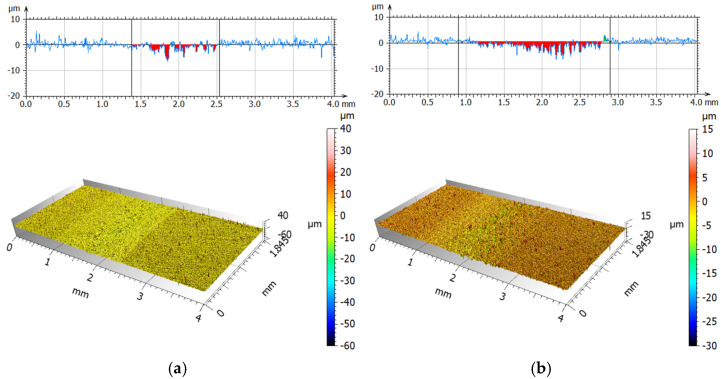
Isometric photos and examples of surface profiles after dry friction (**a**) and with 0.9NaCl; (**b**) for print angle 90° (POM).

**Table 1 materials-19-00132-t001:** Material properties of AIB METAL [24].

Parameter	Test Method	Unit	Value
Tensile strength XY	ISO 527-1	MPa	35 ± 2
Tensile strength ZX	ISO 527-1	MPa	19 ± 2
Elastic modulus	ISO 527-1	GPa	1.5 ± 0.1
Tensile yield strain	ISO 527-1	%	5.1 ± 0.1
Flexural strength	ISO 178	MPa	66 ±- 2
Flexural deflection	ISO 178	mm	9.0 ± - 0.1
Charpy notched impact strength	ISO 179-1	kJ/m^2^	6 ± 1
Density	ISO 1183	g/cm^3^	4
Glass transition temperature	ISO 11357	°C	81

**Table 2 materials-19-00132-t002:** Additive manufacturing parameters [25].

Parameter	Value	Unit
Layer Height	0.12	mm
Wall loops	2	-
Infill pattern	Linear	-
Infill density	100	%
Nozzle temperature	245	°C
Bed temperature	80	°C
Print speed	80	mm/s
First layer height	0.30	mm

**Table 3 materials-19-00132-t003:** Sample designations [25].

Sample Designation	Number of Repetitions	Orientation
S1-1	10	0°
S1-2	10	90°
S2-1	5	0°
S2-2	5	90°

**Table 4 materials-19-00132-t004:** Properties of 100Cr6 steel [26].

Property	Symbol	Unit of Measure	Type	Notes	Values
Density	ρ	g/cm^3^	Physical	Room temp.	7.80
Young’s modulus	E	GPa	Mechanical	-	200
Hardness	H	HRC	Mechanical	-	60–66
Coefficient of linear thermal expansion	α	10^−6^/°C	Thermal	(ΔT = 0–100 °C)	12.3
Thermal conductivity	λ	W/(m·K)	Thermal	Room temp.	42.4
Service temperature	Tserv	°C	Thermal	-	−60 ÷150
Electric resistivity	*ρ*	Ω·m·10^−9^	Electric	-	215
Relative magnetic permeability	µ_m_	-	Magnetic	Ferromagnetic	>300

**Table 5 materials-19-00132-t005:** Properties of POM [27].

Property	Symbol	Unit of Measure	Type	Notes	Values
Density	ρ	g/cm^3^	Physical	Room temp.	1.40
Water absorption	Aw	%	Physical	24 h	0.30
Young’s modulus	E	MPa	Mechanical	-	2800
Friction coefficient	µ	-	Mechanical	Room temp.	0.28
Hardness	H	Shorde D	Mechanical	-	80–90
Compressive yield strength	-	MPa	Mechanical	-	30–120
Coefficient of linear thermal expansion	A	10^−6^/°C	Thermal	(ΔT = 0–100 °C)	93
Thermal conductivity	Λ	W/(m·K)	Thermal	Room temp.	0.27
Service temperature	T_serv_	°C	Thermal	-	−40 ÷ 85
Volume resistivity	*ρ*	Ω·m	Electric	-	>10^13^
Relative magnetic permeability	µ_m_	-	Magnetic	Diamagnetic	<~1

**Table 6 materials-19-00132-t006:** Parameters of tribological test.

Sample—disks	Ø 40 mm—0° Ø 40 mm—90°
Countersample—ball	Ø 6 mm—steel 100Cr6 Ø 6 mm—POM
Type of friction	Rotation
Lubricant	Without lubricant (DF) 0.9NaCl
Radius	14 mm
Load	5 N
Sliding speed	0.1 m/s
Sliding distance	1000 m
Test duration	10,000 s
Humidity	50 ± 5%
Ambient temperature	20 ± 1 °C

**Table 7 materials-19-00132-t007:** Average value of the wear depth and wear area after friction with ball of steel.

Parameters	Value	0° DF	90° DF	0° 0.9NaCl	90° 0.9NaCl
Maximum depth	µm	75.39 ± 2.50	33.01 ± 1.20	79.32 ± 3.20	63.04 ± 2.22
Wear area	µm^2^	56,685 ± 15	16,318 ± 18	57,894 ± 22	39,753 ± 25

**Table 8 materials-19-00132-t008:** Average value of the wear depth and wear area after friction with ball of POM.

Parameters	Value	0° DF	90° DF	0° 0.9NaCl	90° 0.9NaCl
Maximum depth	µm	11.34 ± 1.10	13.87 ± 1.25	7.08 ± 0.75	7.36 ± 0.55
Wear area	µm^2^	2586 ± 26	2476 ± 14	1617 ± 22	3492 ± 25

## Data Availability

The original contributions presented in this study are included in the article. Further inquiries can be directed to the corresponding author.

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
