# Peer review of "Effect of Print Orientation on the Tribological Behavior of a Steel Powder-Modified Thermoplastic"

_materials, 2025, doi:10.3390/ma19010132_

Round 1

Reviewer 1 Report

Comments and Suggestions for Authors

The paper is devoted to an topic of study the samples prepared using Fused Depth Modeling (FDM) 3D printing technology, important for applications and development of the FDN technique. However, the paper need in major revision in points indicated in file attached.

Author Response

Thank you very much for comments.

In some place of the text a short hyphen is used as minus sign or in the range of any parameters, for example, -2 MPa, Refs. [17-19], etc. However, in other text sentences, there is a long minus sign, for example, Refs. [1–3], –60 °C. What the difference? Is meaning of the references ‘[17-19]’ is different in comparison to references [1–3]? Are meanings of the numbers -2 and –2 are different? If they are really different, please describe the differences. If it is the same, then, please, give a long sign of minus in these cases, i.e., –2 MPa, [17–19], etc.

Meaning of these numbers is the same. We changed these signs of minus.

Why the Tables, Figures, and Sections are given in bold through the text in some places? Seems, there is no such requirement in the rules of the journal, excluding the Figure captions and Table heads. Anyway, what the difference between the bold and non-bold names of the Tables and Figures? Please, describe.

There is no difference between bold and non-bold tables and figures. The lack of bolding was simply due to inattention. We sincerely apologize for this.

Text: somewhere in the text the term ‘counter-sample’ is used, in other places ‘countersample’. Please, write rightly and in the same way in the whole text.

We have standardized these issues

Figures 6, 8, 9, 10: The estimated standard deviations should be not only shown graphically but given in the Figs. For example, 24.10(5) and 24.20(5) or 24.10±0.05 and 24.20±0.05 for Fig. 6. It is difficult or there a time necessary for readers to estimate the deviations without the clear indication (especially in cases, if they are shown graphically only). In this case the printing the sentences with the values of the deviations can be cancelled (or can be maintained according of decision of authors).

Thank you for the comment. We added this to the figures.

The values discussed in the text and in Tables in the section of Results should be given with estimated standard deviations or, if the standard deviations are indicated in text, Tables or Figures, at least, should be given down to the last significant digit. For example, for hardness 24.10±0.05, it will be 24.10(5) in a short writing or 24.10, but not 24.1

Standard deviations are included in tables.

2, lines 46 – 47: For convenience of readers, let give not only the abbreviations, but the full names for PLA, PA, and ABC polymers

Ok, thank you. We added explanations to these names.

2, line 69: In spite of 3D is known abbreviation, please give a decryption

Thank you. We changed it.

2,3,4, lines 89 – 91, 134 – 141: Please indicate what % is used, wt. % or at. %.

Thank you for these comments. It is % wt. We write about it in text.

3, Table 1: In Tale 1, please give the units (MPa, GPa, %, etc.) in the first column. Instead +/, please, type ±

Thank you for the comment. These things in table 1 we changed to good version.

line 105: Please, give a decryption of the abbreviation PEI

We gave a decryption of the abbreviation.

Table 3, Figure 1: Table 1 describes samples S1-1,-2 and S2-1,-2. However, the caption to Figure 1 refers to samples S2-1,5,7, S3-1, S2-2,4,6,8 и S3-2. How were crystals S1-1 and S1-2 prepared? The parameters of samples S2-5, -7, S2-4, -6, -8, S3-1,-2 should also be listed in Table 1 if they were used for the experiment (or it should be indicated that they were not studied). Furthermore, the article only indicates which studies used samples S1 and S2. However, the results do not indicate which specific samples were used for the measurements (S1-1, S1-2, S2-1, S2-2, etc.). For each result, obtained by each technique, please indicate which specific sample was used in every subsection of the ResultsSection 3.

I am very sorry. It was our mistake. We deleted samples S-3 in tables but we forgot to do it in text under the figure 1. Now it is everything clear, we hope so.

2 P.6, line 115: Please use the whole name ‘NaCl 0.9 % water solution’ instead of ‘NaCl 0.9 %’. Introduce here the abbreviations DF for ‘dry friction’ and ‘0.9NaCl’ for ‘NaCl 0.9 % water solution’. Use these abbreviations in caption to Figure 4 and in text. Correct the designations ‘0.9 NaCl’ used now in some places in the text (in some places the wright designation ‘0.9NaCl’ is always used).

We correct all of these things.

4, lines 125 – 127 and 144 – 146: Sentence with the name of the counter-samples is printed twice at lines 125 – 127 and 144 – 146. Please cancel the repetition. From this sentence (…samples…were …balls made of 100Cr6 steel (AISI 52100) and an unmodified POM (polyoxymethylene) polymer) follows that the balls contain both steel and polymer together. However, in following text the steel balls and polymer balls are described. Please modify the sentence to separate the balls of different kind.

We write it in this way that we hope it is clear.

5, Tables 4 and 5: Some of the quantities are given without a symbol (designation) or with an unusual symbol. You can use a usual symbols H for hardness and Tserv or Tmeas for service temperature. Usually, symbol ρ, but not δ, is used for the mass density. To distinguish the mass and electrical density, you can use symbols ρmass and ρel. These symbols will be understandable for all readers.

We changed these symbols.

5, line 160 – 162: ‘The hardness of the tested materials was measured using a hardness tester. Figure 2 shows the hardness measurement procedure. The hardness was measured using a Shore hardness tester…’ So, there is twice written that the hardness was measured using a harness tester. Please cancel the repetition.

Ok, we cancelled the repetition.

5, lines 160 – 162, p.6, lines 169 – 170, 175, p.8, line 185: For every device used (harness tester, optical tensiometer, etc.), the producing firm, city and state of the main office of the firm should be indicated.

For every device which we use, we add producing firm, city and country.

7, Table 6: Please indicate that ‘without lubricant’ means ‘DF’ in the table raw name: Without lubricant (DF). Please give the same symbol ‘-‘ or ‘–‘ in the Table and corresponding text and designations. As well, probably, ‘duratiom’ means ‘duration’, please correct.

Thank you for the comment. We corrected all of this.

8, Section 3.1: Using the hardness measurements, the authors have made a conclusion “that the polymer layer has a dominant influence on the mechanical response of the surface. This may indicate that the addition of 316L steel had a smaller effect on the final product than originally expected”. Where these conclusions from? As can understand from the sample preparation and measurement details sections, the samples S1-1 (0° between the sample surface and the build table surface during the printing of the sample) and S1-2 (90°) were prepared from the same composite material containing 316L steel powder embedded in a thermo-83 plastic polymer. There were no samples without 316L steel powder and no measurements of such samples, as well no comparison with any literature data. A conclusion can be drawn about small influence of the sample orientation during the printing but not about the influence of the steel embedding.

Thank you for your comment. We agree that the presented results allow us to draw conclusions regarding the relatively small effect of sample orientation during printing. We have amended the description to reflect the obtained test results. We agree that the effect of steel addition is more complex and cannot be clearly assessed based on the current study. This aspect would require further, detailed investigation and may be addressed in future studies.

9, Section 3.2: From the surface wettability studies, the authors a conclusion drawn that the results indicate the dominance of the polymer phase on the surface without any discussion of proofs. Please provide proofs and a discussion.

We wrote more about proofs and a discussion.

10, 11: Use the decimal points but not commas in the numbers at axes of Figure 9 and 10.

Thank you for the comment. We change commas to points.

10, Section 3.3: It is written in such a way that it is not clear what the conclusions for ball of steels relate to. Either the formation of a thin water layer with ions is considered as a cause of a lower coefficient of friction for the samples with both 0° and 90° printing angles when 0.9NaCl is used or it is concluded that the formation of a thin layer of water with ions using 0.9NaCl 3 leads to a greater decrease in the coefficient of friction (by 16%) than in the case of water (13%). From the sentence which is written as a conclusion it is not possible to understand the relationship. Please write the conclusion clearly.

Thank you for the comment. We added more information about this conclusions.

10, lines 237 – 238: ‘The friction 237 coefficients differed.’ This sentence is written second time. It should be deleted.

Yes, now it is deleted.

11, lines 248, 252, 253, 262: There are new term ‘physiological saline solution’ is used, probably, instead of 0.9NaCl. Please, use the old term (0.9NaCl) or, at least, give it in brackets (physiological saline solution (0.9NaCl)) first time when it appears or give an explanation what is difference.

We use only old version. We hope that now it is clear.

Figures 12 – 14: As can be seen in the articles published in Materials, the designations of the subfigures should be given by bold letters in bold round brackets (i.e., (a), (b), etc.) and placed in middle at bottom. In the caption to the Figure, the designations should be given by bold letters in non- bold round brackets (i.e., (a), (b), etc.). The presentation of Figures should be corrected according to style of the journal.

We corrected presentation of figures to style of the journal.

11,13, Tables 7, 8: Estimated standard deviations should be presented for the measured values in the Table

We added standard devations there.

12, wear test with steel balls: ‘It is also possible that the electrolyte may contribute to tribocorrosion, and consequently, progressive damage to the polymer matrix. The presence of 316L steel particles may result in the formation of local zones with altered mechanical properties and adhesion.’ For which samples these conclusions? For samples prepared with 0° or 90° angle printing or for both? It should be indicated.

It is for both. We wrote this in article.

P 12-13, Conclusions: In conclusion, only the observed results are summarized. This is enough for a laboratory report, but not for a scientific article. In addition to the observed results, the conclusions of a scientific article should contain at least the suggested causes for the observed results and trends in their change with a change in any parameter under study.

We changed it to better.

List of references: The references should be given in a style of Materials. For example, not the whole name of the journal should be given, but a standard abbreviation of the journal. DOI, if exists, should be given. For other requirements, see the instructions for authors.

Thank you for the comment. We add DOIs to all of references and make a standard abbreviation of the journal.

Reviewer 2 Report

Comments and Suggestions for Authors

The manuscript presents a high-quality original contribution to the experimental techniques for characterizing tribological properties of a steel-powder-reinforced thermoplastic composite manufactured by FDM technology. This manuscript certainly fits into the scope of MDPI Materials, as it presents a comprehensive study that effectively links FDM process parameters to material performance. The investigation into a steel-powder-reinforced composite is highly relevant to current materials engineering research. A major strength lies in the multidisciplinary approach, combining mechanical (Shore D), surface (wettability/contact angle), and tribological (ball-on-disc) characterization. The inclusion of two distinct environmental conditions, dry friction and a physiological saline solution, is particularly commendable. This research certainly improved our overall confidence in describing thermoplastic composites with the addition of steel powder obtained by the FDM method, as it provides crucial insight into the material's real-world applicability in potentially biomedical or industrial settings. Literature review is lean and focused. The frameworks for the experimental program and predictive modelling are explained in detail. Discussion is extensive and fair. Overall, the methodology is robust, well-detailed, and promises valuable contributions regarding the anisotropy and wear of FDM-printed composites. The manuscript can certainly be recommended for publication.

Author Response

Thank you for taking the time to review and for all your valuable comments.

Reviewer 3 Report

Comments and Suggestions for Authors

In this study, the authors employed a clear and reproducible experimental methodology with a coherent multi-factor comparative design, an empirical focus on the true anisotropy of FDM parts, and the use of three-dimensional wear analysis. These strengths provide a reliable foundation for analyzing the tribological behavior of 3D-printed composites.

  1. Your title is so long.
  2. In the abstract, FDM is referred to as “Fused Depth Modeling,” whereas the correct term is “Fused Deposition Modeling.”
  3. You should consider an intermediate angle between 0° and 90°, such as 45°.
  4. The number of test repetitions is low.
  5. Why were parameters such as layer height and printing speed kept constant?
  6. You stated that “according to information obtained directly from the manufacturer, the steel content in the material is approximately 95 wt.%,” but the extracted properties are not consistent with this claim.
  7. There are some writing errors.
  8. The wear mechanism was not discussed.
  9. The introduction contains many long and unclear sentences.
  10. Some parameters are explained repeatedly without clear or meaningful distinctions.
  11. It is recommended to use the following paper published in Polymers. 3D and 4D Printing of PETG–ABS–Fe3O4 Nanocomposites with Supreme Remotely Driven Magneto-Thermal Shape-Memory Performance.
  12. The column titles in Tables 7 and 8 are not clear.
  13. The references do not include DOIs.
  14. In Figure 1, S3 is mentioned, but it does not appear in Table 3.
  15. It would be better to consider parameters other than printing direction.
  16. In the friction test, parameters such as environmental humidity were not mentioned.
  17. It would be better to investigate other types of friction, such as their superposition.
  18. Other materials should also be considered.
  19. In some figures, the analysis is incomplete and not sufficiently clear.
  20. The symbol μ is assigned to several different parameters.
  21. There are some spelling errors in the manuscript.
  22. What was the rationale for selecting the parameters listed in Table 6?
  23. A separate section on limitations and future work should be included.
  24. Some figures do not have acceptable quality.
  25. The conclusion is very general.

Author Response

Thank you for the comments.

Your title is so long.

We changed it to shorter. We hope that it is better.

In the abstract, FDM is referred to as “Fused Depth Modeling,” whereas the correct term is “Fused Deposition Modeling.”

Now it is correct.

You should consider an intermediate angle between 0° and 90°, such as 45°.

Thank you for this valuable suggestion. This is true that the investigation of an intermediate angle, such as 45°, could provide additional insight into the studied phenomena. However, this aspect will be considered in future studies.

The number of test repetitions is low.

There are five test repetitions. This is a sufficient number of repetitions to calculate means and standard deviations.

Why were parameters such as layer height and printing speed kept constant?

Thank you for this question. Parameters such as layer height and printing speed were kept constant to minimize their influence on the results and to ensure that the observed differences could be attributed primarily to the variables under investigation. In next article we will try to examine influences of these parameters to tribological properties.

You stated that “according to information obtained directly from the manufacturer, the steel content in the material is approximately 95 wt.%,” but the extracted properties are not consistent with this claim.

Thank you for this comment. The information regarding the steel content (approximately 95 wt.%) was obtained directly from the manufacturer. We acknowledge that the reported material properties may differ from those of bulk steel due to the presence of alloying elements and the specific manufacturing process. It can be reason why the extracted properties are not consistent with this claim. It is the main reason why we tried to check what kind of properties represents this material.

There are some writing errors.

Thank you for the comment. We improve article also in case of writing errors.

The wear mechanism was not discussed.

Wear mechanisms are discussed in a basic manner to provide a general overview of the processes occurring in the system under study. Although the detailed aspects are not fully developed, the information presented allows for drawing appropriate conclusions. Future work could expand the scope of the topics covered to include those mentioned earlier.

The introduction contains many long and unclear sentences.

Thank you for this comment. The Introduction has been revised to improve clarity and readability, and several long sentences have been shortened or rephrased.

Some parameters are explained repeatedly without clear or meaningful distinctions.

Thank you for this comment. Repeated explanations of certain parameters have been reduced or consolidated to improve clarity and avoid redundancy.

It is recommended to use the following paper published in Polymers. 3D and 4D Printing of PETG–ABS–Fe3O4 Nanocomposites with Supreme Remotely Driven Magneto-Thermal Shape-Memory Performance.

We add this article.

The column titles in Tables 7 and 8 are not clear.

We deleted some words from tittle. We hope that now it is better.

The references do not include DOIs.

Thank you for the comment. We add DOIs to all of references.

In Figure 1, S3 is mentioned, but it does not appear in Table 3.

Yes, it is a mistake. We corrected this.

It would be better to consider parameters other than printing direction.

Thank you for this suggestion. We will try to incorporate it into our research for future articles. We know that considering additional parameters beyond print direction can provide further insights.

In the friction test, parameters such as environmental humidity were not mentioned.

Thank you for the comment. We added this information to parameters.

It would be better to investigate other types of friction, such as their superposition.

Thank you for this comment. We agree that investigating other types of friction, including their superposition, would provide additional insight into the tribological behavior of the tested materials. This aspect is planned to be considered in future research.

Other materials should also be considered.

Thank you for this comment. We acknowledge that considering other materials could provide additional insights. However, this aspect is beyond the scope of the present study and it will be probably include in future research.

In some figures, the analysis is incomplete and not sufficiently clear.

We have added longer and more complete analysis to some of the figures.

The symbol μ is assigned to several different parameters.

This symbol in this article means only coefficient of friction.

There are some spelling errors in the manuscript.

Thank you for this comment. The manuscript has been carefully checked and corrected for spelling errors again.

What was the rationale for selecting the parameters listed in Table 6?

The parameters listed in Table 6 were selected to reflect conditions relevant to biomedical tribological systems. There was totally basic tests with these material. A normal load of 5 N was chosen to ensure stable contact conditions while avoiding excessive wear or surface damage. The use of 6 mm balls made of POM and 100Cr6 steel allowed a comparison between polymer–metal and metal–metal contact pairs commonly encountered in biomedical applications. Tests were conducted under dry conditions and in the presence of a 0.9% NaCl solution to simulate both boundary lubrication and physiological environments. This combination of parameters enabled a systematic evaluation of friction and wear mechanisms under conditions representative of practical use. We plan to make friction tests also with other balls or lubrication.

A separate section on limitations and future work should be included.

We added some phrases about these.

Some figures do not have acceptable quality.

We tried to improve quality of these figures. We hope so that now it is better than earlier.

The conclusion is very general.

We changed it to better.

Round 2

Reviewer 1 Report

Comments and Suggestions for Authors

The authors have correct the paper but some points need in further correction or explanation, see file attached

Author Response

Thank you very much for your valuable comments. We tried to include them all.

We hope that the article will be able to be published in its current form.

Reviewer 3 Report

Comments and Suggestions for Authors

The manuscript is well-revised. Accept as is.

Author Response

Thank you.